# Genetic Modifiers and Phenotype of Duchenne Muscular Dystrophy: A Systematic Review and Meta-Analysis

**DOI:** 10.3390/ph14080798

**Published:** 2021-08-13

**Authors:** Carlos Pascual-Morena, Iván Cavero-Redondo, Alicia Saz-Lara, Irene Sequí-Domínguez, Maribel Lucerón-Lucas-Torres, Vicente Martínez-Vizcaíno

**Affiliations:** 1Health and Social Research Center, Universidad de Castilla—La Mancha, 16071 Cuenca, Spain; carlos.pascual@uclm.es (C.P.-M.); ivan.cavero@uclm.es (I.C.-R.); alicia.delsaz@uclm.es (A.S.-L.); mariaisabel.luceron@uclm.es (M.L.-L.-T.); vicente.martinez@uclm.es (V.M.-V.); 2Rehabilitation in Health Research Center (CIRES), Universidad de las Américas, Santiago 72819, Chile; 3Facultad de Ciencias de la Salud, Universidad Autónoma de Chile, Talca 3460000, Chile

**Keywords:** Duchenne muscular dystrophy, polymorphism, TGFβ, SPP1, LTBP4, systematic review, meta-analysis

## Abstract

The transforming growth factor beta (TGFβ) pathway could modulate the Duchenne muscular dystrophy (DMD) phenotype. This meta-analysis aims to estimate the association of genetic variants involved in the TGFβ pathway, including the latent transforming growth factor beta binding protein 4 (*LTBP4*) and secreted phosphoprotein 1 (*SPP1*) genes, among others, with age of loss of ambulation (LoA) and cardiac function in patients with DMD. Meta-analyses were conducted for the hazard ratio (HR) of LoA for each genetic variant. A subgroup analysis was performed in patients treated exclusively with glucocorticoids. Eight studies were included in the systematic review and four in the meta-analyses. The systematic review suggests a protective effect of *LTBP4* haplotype IAAM (recessive model) for LoA. It is also suggested that the *SPP1* rs28357094 genotype G (dominant model) is associated with early LoA in glucocorticoids-treated patients. The meta-analysis of the *LTBP4* haplotype IAAM showed a protective association with LoA, with an HR = 0.78 (95% CI: 0.67–0.90). No association with LoA was observed for the *SPP1* rs28357094. The *LTBP4* haplotype IAAM is associated with a later LoA, especially in the Caucasian population, while the *SPP1* rs28357094 genotype G could be associated with a poor response to glucocorticoids. Future research is suggested for *SPP1* rs11730582, *LTBP4* rs710160, and *THBS1* rs2725797.

## 1. Introduction

Duchenne muscular dystrophy (DMD) is an X-linked recessive lethal disease that predominantly affects males, with an incidence of one case per 3500–9000 live births [1]. It is caused by mutations, insertions, or deletions of the *dystrophin/DMD* gene (location Xp21.2-p21.1) resulting in an absence of the cytoskeletal protein dystrophin. Dystrophin is an actin-binding protein that, through the α/β-dystroglycan complex, links the cytoskeleton to the extracellular matrix protein laminin, stabilizing the surface of muscle cells and preventing their rupture during contraction and relaxation cycles. Therefore, this is essential for the strength, stability, and functionality of the myofibers [2,3]. The absence of dystrophin in the muscle tissue causes a destabilization of the dystrophin-associated glycoprotein complex. This leads to, during contraction and relaxation cycles, sarcolemmal instability, and a decrease in force transmission by the sarcomere occurs. In dystrophic muscle there is an infiltration of mononuclear cells, variation in fiber size, centrally located nuclei, and degeneration and replacement of muscle tissue by fibrotic tissue. This leads to a progressive loss of muscle strength, due to the destruction of muscle tissue, along with the difficulty of muscle contraction of the remaining muscle, due to fibrotic tissue and muscle contractures. Ultimately, loss of ambulation (LoA) occurs in late childhood or early adolescence, in addition to other comorbidities such as cardiac and respiratory failure that reduce quality of life and life expectancy. Despite the optimization of glucocorticoid treatment and other pharmacological and nonpharmacological therapies, the prognosis remains poor [2,3,4,5,6,7,8,9]. However, some variability in the progression of DMD has been observed, even after considering improved treatments.

There are common genetic variants outside of the *dystrophin* gene and that are not pathological, which could influence disease progression through an anti-fibrotic or pro-fibrotic effect. The transforming growth factor beta (TGFβ) pathway is a complex signaling pathway that has been proposed as a candidate for modifying DMD progression, especially the age at LoA and cardiac remodeling. TGFβ is a cytokine that binds to the type II TGFβ receptor, which together with the type I TGFβ receptor forms a heterotetrameric complex, and the type I TGFβ receptor is phosphorylated at a region rich in glycine and serine residues, resulting in activation. Type I TGFβ then phosphorylates certain Smad proteins at C-terminal serine residues in a conserved C-terminal Ser-Ser-X-Ser motif. These phosphorylated Smads oligomerize with Smad4, and these complexes act on target genes, promoting homeostasis, cell differentiation, and tissue regeneration, among others [10]. At the muscle level, TGFβ1 and related cytokines, such as myostatin, a molecule belonging to the TGFβ superfamily, interfere with muscle growth and differentiation factors, such as myoblast determination protein 1 and insulin-like growth factor 2 [11]. This mechanism may also explain the cardiac remodeling that occurs with myocardial fibrosis [12,13].

In *mdx* models, plasma and muscle levels of TGFβ1 and the levels of TGFβ1 and connective tissue growth factor (CTGF) in the sarcoplasm of muscle cells and mesenteric interstitium are increased and correlate with the degree of fibrosis [14]. Two genes whose genetic variants might exert an effect on the phenotype are latent transforming growth factor beta binding protein 4 (*LTBP4*, location 19q13.2) and secreted phosphoprotein 1, also known as osteopontin (*SPP1*, location 4q22.1). LTBP4 binds to TGFβ, reducing its activity [15]. The homozygous IAAM haplotype (V194I, T787A, T820A, and T1140M) might create more stable binding, reducing TGFβ activity [16]. Meanwhile, *SPP1* stimulates TGFβ production and increased inflammatory cell infiltration [15]. Although in *mdx* models, *Spp1* genetic variants resulting in low basal activity are associated with an improved phenotype [17], the opposite result tends to occur in humans, showing the complexity of this pathway. Thus, for *SPP1*, the rs28357094 genetic variant, which is located in the *SPP1* promoter, has been proposed as a possible disease modifier [16].

In 2017, a systematic review suggested the association of some of these genetic variants with LoA [18]. Since then, new studies have been published that may improve the available evidence [19,20,21,22,23,24,25,26]. Therefore, this systematic review and meta-analysis aims to estimate the association of *SPP1* rs28357094, *LTBP4* haplotype IAAM, and other genes directly involved in the TGFβ pathway with the age at LoA and cardiac function in patients with DMD.

## 2. Materials and Methods

This systematic review was conducted according to the Meta-analyses Of Observational Studies in Epidemiology (MOOSE) guidelines [27] and the Cochrane Collaboration Handbook [28], and it was registered in PROSPERO (registration number: CRD42018111191).

### 2.1. Search Strategy

An electronic systematic search was performed in MEDLINE (via PubMed), EMBASE, Web of Science, and Cochrane Library databases from inception until July 2021. Grey literature, including OpenGrey, Theseo, Networked Digital Library of Theses and Dissertations, and Google Scholar, was also searched. The search terms included dystrophy, Duchenne, DMD, Duchenne muscular dystrophy, dystrophinopathy, polymorphism, osteopontin, spp1, latent transforming growth factor, ltbp4, thrombospondin, thbs1, ambulation, loss of ambulation, LoA, cardiomyopathy, ventricular dysfunction, heart failure, cardiac function, and heart function. References included in previous reviews and in the included studies were screened. If necessary, the studies’ authors were contacted. The specific search is described in Appendix B.

### 2.2. Inclusion/Exclusion Criteria

The inclusion criteria were as follows: (1) participants—males with DMD, with no age restrictions; (2) exposure—(i) *LTBP4* haplotypes (i.e., IAAM/IAAM vs. others), occasionally a single SNP may be considered due to linkage disequilibrium; (ii) *SPP1* genetic variants; (iii) other genetic variants in genes directly involved in the TGFβ pathway; (3) outcomes—LoA (main outcome), defined as the age at which the person requires continued use of a wheelchair because of an inability to walk independently, and cardiac function (secondary outcome), including age of onset of dilated cardiomyopathy (DCM) and age of onset of myocardial dysfunction. No language limitations were imposed.

The exclusion criteria were as follows: (1) participants—a population that was not mainly affected by DMD and the effect of genotype on participants with DMD could not be estimated; (2) exposure—genetic variants not directly involved in the TGFβ pathway.

The literature search was conducted independently by two reviewers (CP-M and IC-R), and disagreements were solved by consensus or with a third reviewer (VM-V).

### 2.3. Data Extraction

An ad hoc table was conducted with the following data extracted from the selected studies: (1) reference (authors and publication year), (2) exposure (gene and genetic variant or haplotype), (3) country, (4) sample size, (5) design, and (6) outcomes (LoA and cardiac function). The data extraction was conducted individually by two reviewers (CP-M and IC-R).

### 2.4. Risk of Bias Assessment

To assess the risk of bias, we used the Quality Assessment Tool for Observational Cohort and Cross-Sectional Studies from Study Quality Assessment Tools [29]. This tool has a 14-item checklist. Each item receives a score of good, fair, or poor and includes the following domains: research question, study population, groups recruited from the same population and uniform eligibility criteria, sample size justification, exposure assessed prior to measurement, sufficient timeframe to observe an effect, different levels of the exposure of interest, exposure measures and assessment, repeated exposure assessment, outcome measures, blinding of outcome assessors, follow-up rate, and statistical analysis.

The risk of bias assessment was conducted independently by two reviewers (CP-M and IC-R), and disagreements were solved by consensus or with a third reviewer (VM-V).

### 2.5. Grading the Quality of Evidence

The Grading of Recommendations, Assessment, Development and Evaluation tool (GRADE) was used to assess the strength of the evidence for the main outcome included in the meta-analyses [30]. Depending on the study design, risk of bias, inconsistency, indirect evidence, imprecision, publication bias, larger effect, possible confounding variables, and dose–response gradient, the GRADE tool rates the strength of evidence as high, moderate, low and very low for each intervention and outcome.

### 2.6. Data Synthesis

A narrative synthesis and an ad hoc table were generated for the main outcome. Additionally, forest plots were used to graphically depict the hazard ratio (HR) of LoA with their 95% confidence intervals (95% CIs) for each genetic variant. The forest plots differentiated between patients treated with or without glucocorticoids.

When two or more HRs for the LoA associated with a specific genetic variant and their 95% CI were available, a fixed-effects meta-analysis was performed [31]. If the confidence interval or standard error was not reported, it was calculated from the standard deviation or estimated from the *p*-value [32]. Heterogeneity, as measured by the I^2^ test [28,33], was considered not important <40%, substantial 30–60%, important 50–90%, and considerable >75%. The *p*-value of heterogeneity was also considered to determine whether heterogeneity was significant. The meta-analyses differentiated whether glucocorticoid-treated patients, patients who were not treated with glucocorticoids, or both types of patients were included.

STATA SE software, version 15 (StataCorp, College Station, TX, USA), was used to conduct the statistical analyses.

### 2.7. Infographics

An infographic of the manuscript was created using Canva software (Figure A1).

## 3. Results

Of the 259 studies identified, 8 met the inclusion criteria and were included in the systematic review [19,20,21,22,23,24,25,26], and 4 were included in the meta-analysis (Table 1 and Figure 1) [20,21,22,24]. Four studies were excluded for various reasons (Appendix A) [34,35,36,37].

The main characteristics of the included studies are summarized in Table 1. Three studies used cohorts from the United States [22,25,26], one from China [21], one from the Cooperative International Neuromuscular Research Group (CINRG) [20], one from Italy [19], one from Italy and CINRG [23], and one from France, Italy, the Netherlands and the United Kingdom [24]. A total of 2050 individuals were analyzed, although the use of CINRG cohorts may overrepresent the sample. Five studies analyzed the association of *LTBP4* haplotypes with LoA [19,20,21,22,24], five analyzed the association of *SPP1* genotypes with LoA [19,20,21,23,24], one assessed the interaction and association of *LTBP4* and *THBS1* genotypes with LoA [26], two investigated the association of the *LTBP4* genotype with cardiac function [19,25], and one analyzed the association of the *SPP1* genotype with cardiac function [19].

### 3.1. Loss of Ambulation—Kaplan–Meier Analyses

Table 2 shows the Kaplan–Meier analyses of the included studies. No study showed an association of the *LTBP4* haplotype IAAM (recessive model) vs. other haplotypes with LoA [19,20,21]. *SPP1* rs28357094 showed an association with LoA of 1–1.2 years in favor of the genotype T (recessive model) in two of the three studies [20,23]. Finally, an interaction between *LTBP4* rs710160 genotype CC and *THBS1* rs2725797 genotype TT may result in a protective effect [26].

Regarding the glucocorticoid subgroup, only one study showed an association of *LTBP4* haplotype IAAM vs. other haplotypes, with a difference of 1.8 years in favor of the IAAM haplotype [22]. For *SPP1* rs28357094, one of the two studies showed an association [20], while rs28357094 showed no association in the nonglucocorticoid subgroup [21]. Finally, *SPP1* rs11730582 also showed an association, with the C genotype being protective (dominant model) [21].

### 3.2. Loss of Ambulation—Cox Regression Analyses

Table 3 and Figure 2 show Cox regression analyses of the included studies. The *LTBP4* haplotype IAAM was associated with an HR = 0.77 (*p* = 0.01) and HR = 0.52 (95% CI: 0.34–0.78) compared with other haplotypes [22,24], while in other studies it showed no significant effect [20,21]. *SPP1* rs28357094 showed no association [20,24].

Regarding the glucocorticoid subgroup, the *LTBP4* haplotype IAAM was not associated with HR compared with other haplotypes [20,21]. A comparison of the *SPP1* rs28357094 TT genotype with the GG/GT genotypes revealed an HR = 0.62 (95% CI = 0.42–0.92) [20]. Finally, regarding *SPP1* rs11730582 and rs17524488, only rs11730582 genotype C showed a protective association, with an HR = 0.63 (95% CI: 0.45, 0.89) [21].

### 3.3. Cardiac Function

Table 4 shows Kaplan–Meier analyses of the included studies. *LTBP4* rs10880 was not significantly associated with the age at onset of DCM or myocardial dysfunction [19,25]. However, there was a trend toward a higher percentage of patients without myocardial dysfunction with the rs10880 TT genotype [25]. In the glucocorticoid subgroup, the rs10880 genotype T (recessive model) was associated with a later age of onset of DCM (log-rank *p* < 0.05) [19]. *SPP1* rs28357094 was not associated with the age at onset of DCM or myocardial dysfunction, although the genotype T tended to be harmful [19].

### 3.4. Assessment of the Risk of Bias

The studies fulfilled between 64.3% and 71.4% of the quality criteria proposed by the Quality Assessment Tool for Observational Cohort and Cross-Sectional Studies from Study Quality Assessment Tools. The objective of the study was not correctly specified in 12.5% of the studies. No study justified the sample size used to achieve a specific statistical power, and the authors used the available sample. An evaluation of different exposure levels or their changes over time was not necessary. Finally, blinding of the analysts was not considered necessary in any study. The complete risk assessment is available in Appendix A.

### 3.5. Evidence Assessment

Using the GRADE tool, the *LTBP4* haplotype IAAM/IAAM, *SPP1* rs28357094 TT vs. GG/GT and *LTBP4* haplotype IAAM/IAAM vs. other haplotypes in glucocorticoid-treated patients showed low certainty of evidence. The complete assessment is detailed in Appendix A.

### 3.6. Meta-Analysis

Three meta-analyses were conducted that included the following studies: four studies of *LTBP4* haplotype IAAM compared with other haplotypes in glucocorticoid-treated and nonglucocorticoid-treated patients, two studies of the involvement of *SPP1* rs28357094 in glucocorticoid-treated and nonglucocorticoid-treated patients, and two studies of *LTBP4* haplotype IAAM compared with other haplotypes in patients who were exclusively treated with glucocorticoids. Due to the scarcity of studies, we were unable to conduct meta-analyses of the remaining genotypes or of patients who were not treated with glucocorticoids.

The meta-analysis showed a protective association of the *LTBP4* haplotype IAAM, with an HR = 0.78 (95% CI: 0.67, 0.90) (Figure 3A), while *SPP1* rs28357094 showed no association (Figure 3B). In the glucocorticoid subgroup, no association was observed for the *LTBP4* haplotype IAAM/IAAM (Figure 3C).

Heterogeneity was substantial for the *LTBP4* haplotype IAAM, including glucocorticoid-treated and glucocorticoid-untreated patients, with a heterogeneity of I^2^ = 57.8% (*p* = 0.069). Furthermore, substantial heterogeneity was not observed for *SPP1* rs28357094 in glucocorticoid-treated and glucocorticoid-untreated patients or *LTBP4* in the glucocorticoid subgroup, with I^2^ = 34.5% (*p* = 0.217) and I^2^ = 0.0% (*p* = 0.410), respectively.

## 4. Discussion

### 4.1. Main Findings

This systematic review and meta-analysis provides an overview of the evidence supporting the associations of genetic variants with LoA and cardiac function. Our results suggest a notable effect of genetic variants involved in the TGFβ pathway, especially *LTBP4* haplotype IAAM (in the recessive model), but not in patients who were exclusively treated with glucocorticoids, probably due to confounding factors. *SPP1* rs28357094 did not display a significant association. However, the use of glucocorticoids by patients carrying the *SPP1* rs28357094 genetic variant potentially increased the association in favor of genotype T (recessive model). Further research on *THBS1* genetic variants is needed, and the limited evidence available indicates an interaction with *LTBP4* that might improve the DMD phenotype. Finally, more research is needed on other proinflammatory and profibrotic pathways that might exert some effect and distort the effect observed in our study.

### 4.2. Interpretation

In dystrophic mice there is a 36-base-pair insertion/deletion site in a proline-rich domain of *Ltbp4*. The 12 amino acid insertion confers resistance to proteolysis of the TGFβ–LTBP4 complex, reducing TGFβ activity and therefore muscle fibrosis. Although this indel is not present in humans, the haplotypes mentioned in the previous sections (i.e., IAAM and VTTT) represent more than 80% of the total of all possible combinations in the human population. The haplotype IAAM behaves like the 36-base-pair insertion in *Ltbp4* of dystrophic mice, giving LTBP4 increased binding avidity to TGFβ, thereby reducing its activity [38]. Our meta-analysis shows that the *LTBP4* haplotype IAAM (or the rs10880 genotype T) is a protective factor resulting in prolonged ambulation. However, in patients treated exclusively with glucocorticoids, the haplotype IAAM had no effect, probably due to the inclusion of only two studies, one including patients from China, in which the *LTBP4* haplotype IAAM does not appear to be associated with LoA, perhaps due to the genetic/ethnic factors described below. Considering the Kaplan–Meier analyses, Barp A et al. and Chen M et al. did not observe an association, while Bello L et al. and Flanigan KM et al. reported a trend toward benefit for the haplotype IAAM, especially in the glucocorticoid-treated population. Interestingly, the cohorts in the former two studies (Barp A et al. and Chen M et al.) lost ambulation earlier than the cohorts in the latter two studies (Bello L et al. and Flanigan KM et al.), which might imply differences in the care and management or genetic/ethnicity of these patients. Thus, in the cohort analyzed by Barp A et al., the use of glucocorticoids was still not a universal standard, and a relatively low use of these drugs was reported. *Chen M* et al. investigated Asian cohorts, in which the effect of the *LTBP4* haplotype IAAM was different than that on Caucasian cohorts. Furthermore, previous evidence [38] has identified rs710160 as a genetic variant that may modulate the association obtained for the *LTBP4* haplotype IAAM with LoA. Thus, rs710160 genotype C together with the IAAM haplotype leads to less profibrotic signaling, potentially resulting in milder DMD. Additionally, rs710160 has a significant linkage disequilibrium with rs10880 in Caucasian populations but not in other populations, which might explain why the haplotype IAAM has a higher association with LoA in the Caucasian population but not in the Asian population, as described in the study by Chen M et al.

Thrombospondin-1, encoded by the *THBS1* gene (location 15q14), is a potent activator of the latent TGFβ complex. To activate TGF-β1, thrombospondin-1 interacts with the N-terminal region of the latency-associated protein, which binds non-covalently to TGF-β1. Thus, a trimolecular complex is formed, a conformational change occurs, and the reactivity of TGF-β1 is altered. TGF-β1 activation by thrombospondin-1 may be essential for the development of the heart, liver, bones, testes, and hematopoietic systems, among other tissues and organs [39]. Moreover, it exerts an anti-angiogenic effect by decreasing the activities of the nitric oxide and vascular endothelial growth factor (VEGF) pathway, and blocking endothelial cell migration, [39,40], which is harmful in animal models [41]. Finally, thrombospondin-1 could be useful in maintaining low but constant levels of active TGF-β1 [39]. In our study, *THBS1* rs2725797 genotype T is associated with lower thrombospondin-1 activity, with an additive effect with *LTBP4* rs10880 allele C, which may improve the phenotype of the disease, as suggested by the data presented in this review.

SPP1 is a cytokine secreted by macrophages and myoblasts. It belongs to the family of small integrin-binding ligand N-linked glycoprotein secreted phosphoproteins and is expressed in numerous tissues in response to tissue damage/regeneration and inflammatory response. Alternative splicing and post-translational modifications make it a difficult cytokine to study. In *mdx* models without osteopontin, mice had reduced TGFβ levels, fibrosis, and increased strength, with reduced infiltration of inflammatory cells, neutrophils, and natural-killer T cells and increased numbers of regulatory T cells and M2 macrophages. Although the reduction in osteopontin could delay regeneration in acute damage, in mdx models (and probably in DMD) it decreases fibrosis caused by chronic damage [38,42]. Although in our study *SPP1* rs28357094 was not associated with LoA in a population that was not stratified by glucocorticoid use, in the glucocorticoid subgroup, it appears to exert a glucocorticoid-dependent effect on LoA in regression analyses. In fact, in subgroup analyses stratified by glucocorticoid treatment, this genetic variant showed no association with LoA in the glucocorticoid-untreated subgroup or when a low percentage of glucocorticoid-treated patients was included. *SPP1* rs28357094 genotype G was detrimental to patients using glucocorticoids, with earlier LoA. Allele G is associated with lower osteopontin transcriptional activity under basal conditions [43]. However, *SPP1* likely contains glucocorticoid receptor elements, which modulate osteopontin expression through interactions with NF-kB, estrogens, and glucocorticoids. Thus, the *SPP1* rs28357094 genotype G might increase osteopontin expression by 3-fold in the presence of glucocorticoids, increasing profibrotic signaling [44]. *SPP1* rs11730582 might also use a similar mechanism. Genotype C, which results in higher initial osteopontin levels [45], is significantly associated with a later LoA in glucocorticoid-treated patients, suggesting that its effect is glucocorticoid-dependent, similar to rs28357094. Therefore, *SPP1* rs28357094 and perhaps rs11730582 are postulated to function not as disease modifiers but as predictors of a good or poor response to glucocorticoid treatment.

Regarding cardiac complications, no clear association with genetic variants can be observed due to the few included studies. *LTBP4* rs10880 genotype T tended to delay the age of DCM onset, with a significant association in patients treated with glucocorticoids [19]. This result is consistent with the data obtained for LoA. This association may be due to an additive effect of the *LTBP4* haplotype IAAM and glucocorticoids, but this requires further research. In contrast, another study did not detect an association of *LTBP4* with the development of left ventricular dysfunction. Interestingly, it was found that, among patients without left ventricular dysfunction, there was a higher proportion of patients with the rs10880 genotype T [25]. These discrepancies are, perhaps, due to the relatively small sample size, the age of the cohorts, or the categorization of patients into glucocorticoid-treated/untreated groups. Furthermore, the *SPP1* rs28357094 genotype T tended to be harmful in the development of DCM, including in the glucocorticoid subgroup [19]. This fact is interesting, since it is in the opposite direction to what happens in skeletal muscle. Conversely, it is in agreement with what has been observed in animal models, in which an overexpression of *Spp1* causes myocarditis and myocardial dilation [46]. Future research is needed in this regard. Cardiac complications, and especially heart failure, are one of the leading causes of death in patients with DMD. Both the *SPP1* genotype and glucocorticoids use could have some negative impact on the development of dilated cardiomyopathy. However, it should be noted that this does not necessarily imply progression to heart failure, and especially glucocorticoids could delay the onset of heart failure through other mechanisms.

The importance of the TGFβ pathway in the DMD phenotype may have several clinical implications: first, drug development aimed at the downregulation of TGFβ signaling to reduce muscle fibrosis. This approach includes angiotensin 1–7, halofuginone, anti-TGFβ1 antibodies, ixazomib, and angiotensin-II type 1 receptor blockers, which antagonize or downmodulate the TGFβ pathway with promising results in animal models [47,48,49,50,51]. However, some of these drugs have produced undesirable pleiotropic effects, which might be a problem in achieving a clinical benefit [49]. Another pathway is myostatin inhibition using compounds such as follistatin, ACE-031, domagrozumab, and the GDF11 propeptide that inhibit or antagonize the effect of myostatin [52,53,54,55,56,57]. However, the results from animal models and patients with DMD have raised some concerns related to other biological functions of myostatin that affect the metabolism and oxidative capacity of muscle fibers [52,58]. Second, genotyping of patients with DMD for *LTBP4* and *SPP1* could be considered, especially in the Caucasian population. Although knowledge of the genotype of *LTBP4* haplotype/rs10880 and *SPP1* rs28357094 would not alter the medical treatment of the patient, it can potentially provide the clinician and the patient and their family with more individualized prognosis as to their possible evolution and expected response to glucocorticoids. Third, genotyping can be considered in clinical trials of new drugs, such as treatments designed to restore dystrophin expression. Perhaps, by subgroup analysis according to genotype, part of the variability found in the results could be explained. It is possible that the patient’s genotype can determine the response to small changes in dystrophin expression. Finally, the study of genetic variants in unknown DMD patients could be very interesting. Clinical DMD is well established, with diagnosis in early childhood. However, there are rare cases of dystrophinopathies, including Becker muscular dystrophy (BMD) and DMD, that remain undiagnosed until their debut in adolescence or adulthood, mainly due to cardiac involvement. [59,60,61,62,63]. Moreover, there are phenotypically intermediate forms of diagnosed dystrophinopathies, with slowly evolving DMD or rapidly evolving BMD [38]. These exceptional cases could be an opportunity to study the effect of genetic factors (known and unknown) as well as environmental factors on the progression of DMD and other dystrophinopathies.

### 4.3. Limitations

Some limitations should be acknowledged. First, the scarcity of studies included in the meta-analyses limits the statistical power and external validity of the results in larger cohorts of patients with DMD. Second, due to the limited number of studies, we were unable to perform publication bias, metaregression, or sensitivity analyses. This could question the association for *LTBP4* and LoA. In systematic reviews, it is not uncommon for the first published studies to show stronger associations than subsequent ones, and they are even less likely to be published unless they question previous findings. Third, in some studies, the *LTBP4* haplotype was determined using a single SNP (mainly rs10880). Despite the large linkage disequilibrium, it could slightly underestimate the observed result. Fourth, in the Kaplan–Meier survival analysis, we were not always able to consider ethnic differences, the effects of other genetic variants, or a different proportion of glucocorticoid-treated patients, which would potentially modify the observed association. Fifth, participants who are candidates for exon 8 or 44 skipping tend to have slower disease progression [64,65]. Most studies did not consider this possible confounding genotype, and although it is possible that there are no statistically significant differences in the prevalence of these genotypes in the included studies, it cannot be ruled out, considering the ethnic and geographical diversity of the participants, overestimating or underestimating the effect observed for the genetic variants studied. Sixth, other genetic variants, such as *LTBP4* rs710160, and their possible differences in prevalence in different populations, might modify the association expected for the *LTBP4* haplotype IAAM. Seventh, some confidence intervals of hazard ratios were estimated from the *p*-value, which might differ from the true confidence interval. Eighth, in general, the authors did not determine the level of fibrosis of the patients’ muscle tissue, which, in theory, should correlate negatively with LoA or cardiac function. This is probably due to the complexity of the procedures to obtain this information.

## 5. Conclusions

Some of the variability in the DMD phenotype can be explained by the genetic variants directly involved in the TGFβ pathway. Thus, the *LTBP4* haplotype IAAM may predict a subsequent LoA, independent of glucocorticoid use, although glucocorticoids may exert an additive effect on this outcome. This haplotype might also be associated with better cardiac function, although more research is needed to confirm or reject this potential association. *SPP1* rs28357094 and possibly rs11730582 exert a glucocorticoid-dependent effect, with the rs28357094 genotype G serving as a predictor of a poor response. Further research is needed to establish the possible interaction of *LTBP4* rs710160 and *THBS1* rs2725797. Moreover, future studies for the *LTBP4* haplotype IAAM in non-Caucasian cohorts are required to confirm the findings in this review, as well as the possibility that the association of the *LTBP4* haplotype IAAM and LoA is conditioned by the genetic variant rs710160, being relevant in non-Caucasian populations. Knowledge of the effect of the TGFβ pathway on the DMD phenotype provides a possibility of developing new treatments, and genotyping of the genetic variants involved could provide the clinician and the patient with more information on their possible progression or response to certain pharmacological treatments.

## Figures and Tables

**Figure 1 pharmaceuticals-14-00798-f001:**
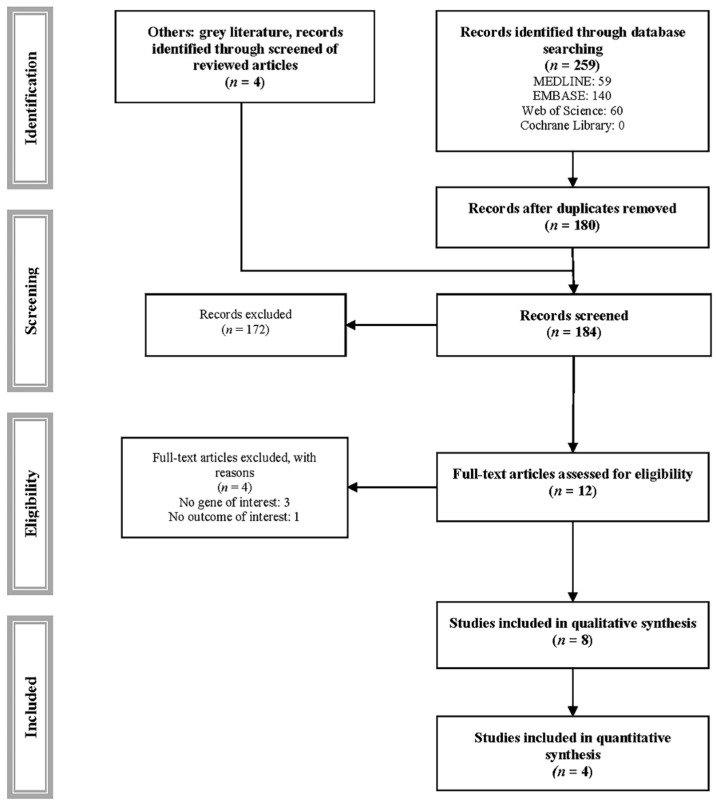
PRISMA flowchart of study selection.

**Figure 2 pharmaceuticals-14-00798-f002:**
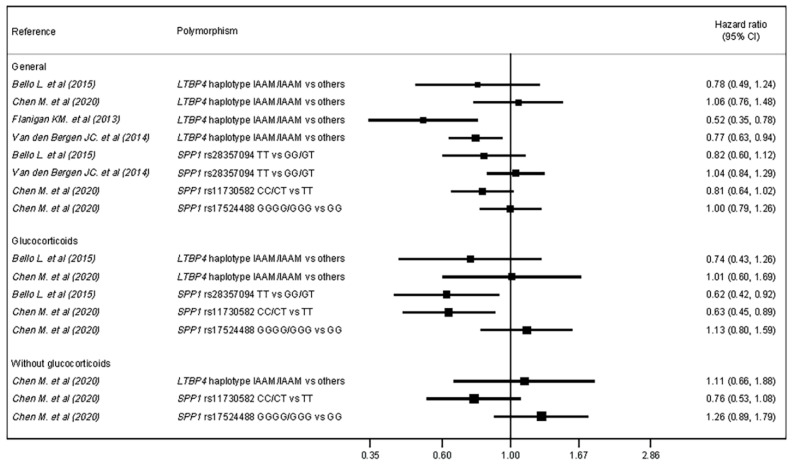
Forest plot for hazard ratio of LoA by genetic variant (polymorphism) and glucocorticoid subgroup.

**Figure 3 pharmaceuticals-14-00798-f003:**
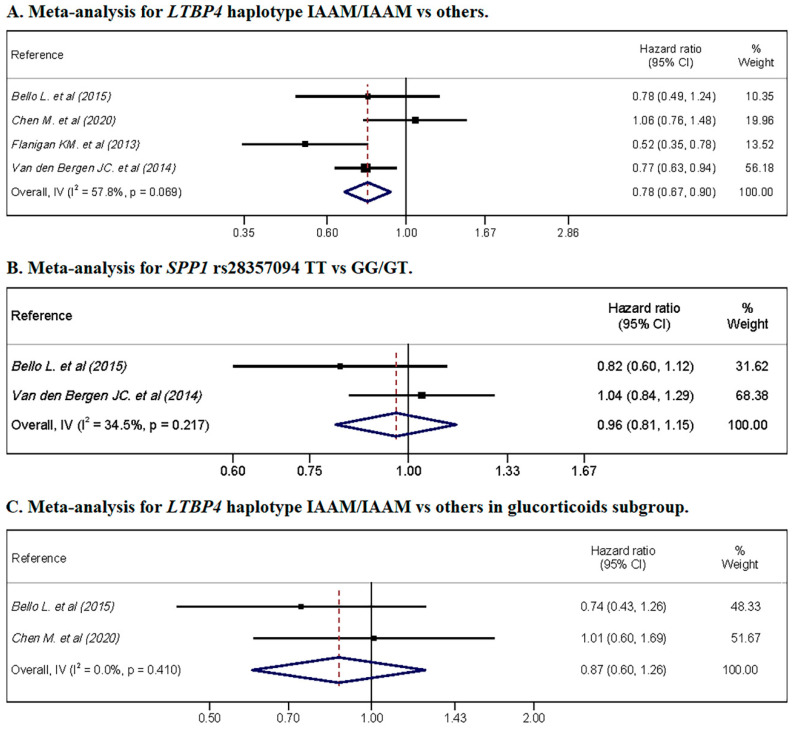
Meta-analyses for hazard ratio of LoA by genetic variant. (**A**) shows the pooled for *LTBP4* haplotype IAAM/IAAM vs. others, without glucocorticoids restrictions; (**B**) shows the pooled for *SPP1* rs28357094 genotype TT vs. GG/GT, without glucocorticoids restrictions; (**C**) shows the forest plot for *LTBP4* haplotype IAAM/IAAM vs. others in glucocorticoids-treated cohorts.

**Table 1 pharmaceuticals-14-00798-t001:** Characteristics of the included studies.

Reference	Country	Gene	Genetic Variant	Sample	Design	Outcomes
Barp A. et al. (2015) [19]	Italy	*SPP1* *LTBP4*	rs28357094 rs10880	178	Retrospective	CF LoA
Bello L. et al. (2015) [20]	CINRG	*SPP1* *LTBP4*	rs28357094 rs10880	340	Prospective	LoA
Chen M. et al. (2020) [21]	China	*SPP1* *LTBP4*	rs28357094, rs11730582, and rs17524488 rs2303729, rs1131620, rs1051303, and rs10880	326	Retrospective	LoA
Flanigan KM. et al. (2013) [22]	US	*LTBP4*	Haplotype IAAM/IAAM vs. others	254	Retrospective	LoA
Pegoraro E. et al. (2011) [23]	Italy CINRG	*SPP1*	rs28357094	262	Retrospective	LoA
Van den Bergen JC. et al. (2015) [24]	France Italy The Netherlands UK	*SPP1* *LTBP4*	rs28357094 Haplotype IAAM/IAAM vs. others	336	Retrospective	LoA
Van Dorn CS et al. (2018) [25]	US	*LTBP4*	rs10880	101	Retrospective	CF
Weiss RB. et al. (2018) [26]	US	*LTBP4 + THBS1*	rs710160 rs2725797	253	Retrospective	LoA

CINRG = Cooperative International Neuromuscular Research Group; CF: Cardiac function LoA = Loss of ambulation.

**Table 2 pharmaceuticals-14-00798-t002:** Studies including as outcome age of loss of ambulation (Kaplan–Meier analyses).

Authors	Genetic Variant	Subgroup	Loss of Ambulation	
			Effect Size	Statistical Significance
Barp A. et al. (2015) [19]	*SPP1* rs28357094	Total	LoA_TT_ = 10.0y.; LoA_GG/GT_ = 10.5y.	*p* = ns
		Glucocorticoids	LoA_TT_ = 11.3y.; LoA_GG/GT_ = 10.9y.	*p* = ns
		No glucocorticoids	LoA_TT_ = 9.9y.; LoA_GG/GT_ = 10.3y.	*p* = ns
	*LTBP4* rs10880	Total	LoA_TT_ = 9.9y.; LoA_CC/CT_ = 10.9y.	Log-rank *p* = 0.058
		Glucocorticoids	LoA_TT_ = 10.9y.; LoA_CC/CT_ = 11.9y.	*p* = ns
		No glucocorticoids	LoA_TT_ = 9.9y.; LoA_CC/CT_ = 9.9y.	*p* = ns
Bello L. et al. (2015) [20]	*SPP1* rs28357094	Total	LoA_TT_ = 13.0y; LoA_GG/GT_ = 11.8y.	Log-rank *p* = 0.048
		Glucocorticoids	LoA_TT_ = 13.9y.; LoA_GG/GT_ = 12.0y	Log-rank *p* = 0.032
		No glucocorticoids	LoA_TT_ = 10.0y.; LoA_GG/GT_ = 10.0y	Log-rank *p* = 0.6
	*LTBP4* rs10880	Total	LoA_TT_ = 13.9y.; LoA_CC/CT_ = 12.0y.	Log-rank *p* = 0.20
		Glucocorticoids	LoA_TT_ = 13.9y.; LoA_CC/CT_ = 13.3y.	Log-rank *p* = 0.27
		No glucocorticoids	LoA_TT_ = 9.1y.; LoA_CC/CT_ = 10.0y	NA
Chen M. et al. (2020) [21]	*SPP1* rs11730582	Total	LoA_CC/CT_ = 11.00y.: LoA_TT_ = 10.33y.	Log-rank *p* = 0.272
		Glucocorticoids †	LoA_CC/CT_ = 12.00y.; LoA_TT_ = 10.67y	Log-rank *p* = 0.006
		No glucocorticoids †	LoA_CC/CT_ = 9.92y.; LoA_TT_ = 9.33y.	Log-rank *p* = 0.104
	*SPP1* rs17524488	Total	LoA_GGGG/GGG_ = 10.50y.; LoA_GG_ = 10.67	Log-rank *p* = 0.983
		Glucocorticoids †	LoA_GGGG/GGG_ = 11.42y.; LoA_GG_ = 11.92y.	Log-rank *p* = 0.478
		No glucocorticoids †	LoA_GGGG/GGG_ = 9.50y.; LoA_GG_ = 10.00y.	Log-rank *p* = 0.173
	*LTPB4* IAAM/IAAM vs. others	Total	LoA_IAAM/IAAM_ = 10.50y.; LoA_others_ = 10.50y.	Log-rank *p* = 0.706
		Glucocorticoids †	LoA_IAAM/IAAM_ = 10.67y.; LoA_others_ = 11.58y.	Log-rank *p* = 0.960
		No glucocorticoids †	LoA_IAAM/IAAM_ = 9.92y.; LoA_others_ = 9.83y.	Log-rank *p* = 0.676
Flanigan KM. et al. (2013) [22]	*LTBP4* IAAM/IAAM vs. others	Glucocorticoids	LoA_IAAM/IAAM_ = 12.5y.; LoA_others_ = 10.7y.	SD_IAAM/IAAM_ = 3.3y.; SD_others_ = 2.1y.
		No glucocorticoids	LoA_IAAM/IAAM_ = 11.2y.; LoA_others_ = 9.8y.	SD_IAAM/IAAM_ = 2.7y. SD_others_ = 2.0y.
Pegoraro E. et al. (2011) [23]	*SPP1* rs28357094	Total	Group GG/GT earlier loss of ambulation that TT At 14 years, 20% TT were ambulant, nobody of GG/GT	p_kaplan meyer_ = 0.035
Weiss RB. et al. (2018) [26]	*LTBP4* rs710160 and *THBS1* rs2725797	Total	Interactions between *LTBP4* and *THBS1* (∆means) *LTBP4* rs710160 TT + *THBS1* rs2725797 TT: ∆LoA = 0y. *LTBP4* rs710160 CC + *THBS1* rs2725797 CC: ∆LoA = 1.2y. *LTBP4* rs710160 CC + *THBS1* rs2725797 TT: ∆LoA = 6.8y.	NA

LoA measured as medians, unless otherwise specified; † Only truncated mutations, as defined by the authors; ns: not statistically significant; NA = Not available.

**Table 3 pharmaceuticals-14-00798-t003:** Studies including as outcome age of loss of ambulation (Cox regression analyses).

Authors	Genetic Variant	Subgroup	Hazard Ratio for LoA	
			Effect Size	Statistical Significance
Bello L. et al. (2015) [20]	*SPP1* rs28357094	Total	HR = 0.82	95% CI = 0.59–1.12
		Glucocorticoids	HR = 0.62	95% CI = 0.42–0.92
	*LTBP4* rs10880	Total	HR = 0.78	95% CI = 0.49–1.24
		Glucocorticoids	HR = 0.74	95% CI = 0.44–1.26
Chen M. et al. (2020) [21]	*SPP1* rs11730582	Total	HR = 0.81	95% CI = 0.64–1.02
		Glucocorticoids †	HR = 0.63	95% CI = 0.45–0.89
		No glucocorticoids †	HR = 0.76	95% CI = 0.53–1.08
	*SPP1* rs17524488	Total	HR = 1.00	95% CI = 0.80–1.26
		Glucocorticoids †	HR = 1.13	95% CI = 0.80–1.59
		No glucocorticoids †	HR = 1.26	95% CI = 0.89–1.79
	*LTPB4* IAAM/IAAM vs. others	Total	HR = 1.06	95% CI = 0.76–1.48
		Glucocorticoids †	HR = 1.01	95% CI = 0.61–1.69
		No glucocorticoids †	HR = 1.11	95% CI = 0.66–1.88
Flanigan KM. et al. (2013) [22]	*LTBP4* IAAM/IAAM vs. others	Total	HR = 0.52	95% CI = 0.34–0.78
Van den Bergen JC. et al. (2015) [24]	*SPP1* rs28357094	Total	HR = 1.04	*p* = 0.73
	*LTBP4* IAAM/IAAM vs. others	Total	HR = 0.77	*p* = 0.01

† Only truncated mutations, as defined by the authors.

**Table 4 pharmaceuticals-14-00798-t004:** Studies including cardiac function as an outcome.

Reference	Genetic Variant	Subgroup	Cardiac Function
Barp A. et al. (2015) [19]	*SPP1* rs28357094	Total	DCM_TT_ = 19.1y. DCM_GG/GT_ = 24.1y. (*p* = ns)
		Glucocorticoids	DCM_TT_ = 17.0y. DCM_GG/GT_ = 24.0y. (*p* = ns)
	*LTBP4* rs10880	Total	DCM_TT_ = 29.5y. DCM_CC/CT_ = 19.0y. (Log-rank *p* = 0.13)
		Glucocorticoids	DCM_TT_ = >50% without DCM in the end DCM_CC/CT_ = 17.9y. (Log-rank *p* < 0.05)
Van Dorn CS et al. (2018) [25]	*LTBP4* rs10880	Total	Myocardial dysfunction: CC (N = 20): 14.5 ± 3.2y. CT (N = 12): 13.1 ± 3.2y. TT (N = 2): 11.0 ± 2.8y. *p* = 0.21

DCM = Dilated cardiomyopathy; ns: not statistically significant.

## Data Availability

The datasets generated and analyzed are available from the corresponding author.

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
