# Peer review of "Genetic Modifiers and Phenotype of Duchenne Muscular Dystrophy: A Systematic Review and Meta-Analysis"

_pharmaceuticals, 2021, doi:10.3390/ph14080798_

Round 1
Reviewer 1 Report
The manuscript entitled " Association of genetic polymorphisms with the phenotype of 2 Duchenne muscular dystrophy: A systematic review and meta-3 analysis" by Pascual- Morena et al., performed an interesting meta-analysis of two genes of the TGF-beta signaling pathway in the DMD, evaluating whether the polymorphisms of LTB4 and SPP1 would alter the progression of the disease. The study is well designed, using correct statistical analyses and carefully proposing a relationship between the polymorphisms of LTB4 and SPP1 with DMD.
There is only a minor comment that the Authors should address:
Line 71: The Authors should include the studies mentioned in the sentence.
Author Response
- Line 71: The Authors should include the studies mentioned in the sentence.
Authors
Thank you. Done.
Reviewer 2 Report
The authors have performed a retrospective study. They have studied the association between the polymorphisms in LTBP4 and SPP1 genes related to the TGFβ pathway and DMD patients - loss of ambulation (LoA) and cardiac function. Interestingly, they observed an association. The major limitation of the study is low number of sample size ( e.g. 8 for systematic review and 4 for meta analysis. My comments are provided below
- My major concern is whether these polymorphisms can actually be confirmed in the unknown DMD patients. The authors should discuss about this point in the discussion section.
- The authors should discuss about the role of dystrophin in the introduction section.
- The author should describe about the cellular localization of these two genes in the introduction section.
- Did the author notice whether the polymorphism in these two genes are specific only to DMD?
- The authors should mention about the level of fibrosis in these patients.
- The major cause of death in DMD is due to loss of cardiac function in the late stage of disease. Did the author observe any significant correlation between the polymorphism and age as they have mentioned about it in Table 4?
Author Response
- My major concern is whether these polymorphisms can actually be confirmed in the unknown DMD patients. The authors should discuss about this point in the discussion section.
Authors
The reviewer’s comment seems judicious. As suggested, we have included information on this point in the discussion section as follows (page 13):
“Finally, the study of genetic variants in unknow DMD patients could be very interesting. Clinical DMD is well established, with diagnosis in early childhood. However, there are rare cases of dystrophinopathies, including Becker muscular dystrophy (BMD) and DMD, that remain undiagnosed until their debut in adolescence or adulthood, mainly due to cardiac involvement. [55-59]. Moreover, there are phenotypically intermediate forms of diagnosed dystrophinopathies, with slowly evolving DMD, or rapidly evolving BMD [34]. These exceptional cases could be an opportunity to study the effect of genetic factors (known and unknown) as well as environmental factors on the progression of DMD and other dystrophinopathies.”
- The authors should discuss about the role of dystrophin in the introduction section.
Authors
Thank you for the reviewer's comment. As suggested, we have included information on the role of dystrophin in the introduction section (page 2).
- The author should describe about the cellular localization of these two genes in the introduction section.
Authors
Thank you for the reviewer's comment. We have included the location of the LTBP4 and SPP1 genes in the introduction section.
- Did the author notice whether the polymorphism in these two genes are specific only to DMD?
Authors
Thank you for the reviewer's comment. We have clarified this aspect in the introduction as follows (page 2).
“There are common genetic variants outside of the dystrophin gene and that are not pathological, which could influence disease progression through an anti-fibrotic or pro-fibrotic effect.”
- The authors should mention about the level of fibrosis in these patients.
Authors
The reviewer’s comment seems judicious. The authors did not report this information. Therefore, we have included it in the limitations section as follows (page 13):
“Eighth, in general, the authors did not determine the level of fibrosis of the patients' muscle tissue, which, in theory, should correlate negatively with LoA or cardiac function. This is probably due to the complexity of the procedures to obtain this information.”
- The major cause of death in DMD is due to loss of cardiac function in the late stage of disease. Did the author observe any significant correlation between the polymorphism and age as they have mentioned about it in Table 4?
Authors
The reviewer’s comment seems judicious. Due to the small number of studies, no clear association can be observed. We have raised some hypotheses that we have described in more detail in the discussion section (page 12):
“Regarding cardiac complications, no clear association with genetic variants can be observed due to the few included studies. LTBP4 rs10880 genotype T tended to delay the age of DCM onset, with a significant association in patients treated with glucocorticoids [19]. This result is consistent with the data obtained for LoA. This association may be due to an additive effect of the LTBP4 haplotype IAAM and glucocorticoids, but requires further research. In contrast, another study did not detect an association of LTBP4 with the development of left ventricular dysfunction. Interestingly, it was found that, among patients without left ventricular dysfunction, there was a higher proportion of patients with the rs10880 genotype T [25]. These discrepancies are, perhaps, due to the relatively small sample size, the age of the cohorts, or the categorization of patients into gluco-corticoid-treated/untreated groups. Furthermore, the SPP1 rs28357094 genotype T tended to be harmful in the development of DCM, including in the glucocorticoid sub-group [19]. This fact is interesting, since it is in the opposite direction to what happens in skeletal muscle. Conversely, it is in agreement with what has been observed in animal models, in which an overexpression of Spp1 causes myocarditis and myocardial dilation [42]. Future research is needed in this regard. Cardiac complications, and especially heart failure, are one of the leading causes of death in patients with DMD. Both SPP1 genotype and glucocorticoids use could have some negative impact on the development of dilated cardiomyopathy. However, it should be noted that this does not necessarily imply progression to heart failure, and especially glucocorticoids could delay the onset of heart failure through other mechanisms.”
Reviewer 3 Report
The authors present an interesting article reviewing information about GENETIC MODIFIERS of the DMD phenotype rather than genetic polymorphisms associated with the DMD phenotype.
As suggestions "genetic modifiers" instead of "Association of genetic polymorphisms" would bring a more precise idea of the work described in the paper. This is a major change since DMD is a monogenic disease, is not multifactorial such as cancer or lupus. The aspect that changes is disease progression or response to treatment due to other genes related to muscular physiology.
The concept "Genetic variants" instead of polymorphisms is also preferred.
A more focused explanation with regard to muscular physiology and genetic modifiers is required.
It is interesting that authors selected outcomes such as Lost of ambulation, cardiac function, steroid use, or not. This is a positive aspect of the work.
An infographic is desirable to make the paper more visually attractive
Author Response
- As suggestions "genetic modifiers" instead of "Association of genetic polymorphisms" would bring a more precise idea of the work described in the paper. This is a major change since DMD is a monogenic disease, is not multifactorial such as cancer or lupus. The aspect that changes is disease progression or response to treatment due to other genes related to muscular physiology.
Authors
Thank you. As suggested, we have modified the title:
“Genetic modifiers and phenotype of Duchenne muscular dystrophy: A systematic review and meta-analysis”
- The concept "Genetic variants" instead of polymorphisms is also preferred.
Authors
Thank you. As suggested, we have modified throughout the manuscript.
- A more focused explanation with regard to muscular physiology and genetic modifiers is required.
Authors
Thank you for the reviewer’s comment. As suggested, we have expanded on the information regarding the biological and modulatory role in DMD of SPP1, LTBP4 and THBS1 in the discussion section (page 11 and 12).
- An infographic is desirable to make the paper more visually attractive
Authors
Thank you for the reviewer's comment. We have included an infographic of the manuscript in Appendix B:
“2.7. Infographics
An infographic of the manuscript was performed using Canva software (Appendix B).”
Round 2
Reviewer 2 Report
The authors have addressed my concerns. Now the quality of the manuscript has increased significantly. I support the publication of the manuscript.